# The Modification of the Dynamic Behaviour of the Cyclonic Flow in a Hydrocyclone under Surging Conditions

Muaaz Bhamjee *⬤, Simon H. Connell ⬤ and André Leon Nel ⬤

Department of Mechanical Engineering Science, Faculty of Engineering and the Built Environment, University of Johannesburg, Kingsway Road, Auckland Park, Johannesburg 2006, Gauteng, South Africa
* Correspondence: muaazb@uj.ac.za; Tel.: +27-11-559-3476

**Abstract:** The aim in this study was to determine how surging modifies the dynamic behaviour of the cyclonic flow in a hydrocyclone using computational fluid and granular dynamics models. The Volume-of-Fluid model was used to model the air-core formation. Fluid–particle, particle–particle, and particle–wall interactions were modelled using an unsteady two-way coupled Discrete Element Method. Turbulence was modelled using both the Reynold's Stress Model and the Large Eddy Simulation. The model predictions indicate that the phenomenon of surging modifies the dynamics of the cyclonic flow in hydrocyclones and subsequently impacts separation. The results reveal that the primary cyclonic separation mechanisms break down during surging and result in air-core suppression. The flow and primary separation mechanism in the core of the hydrocyclone is driven by the pressure drop and the flow and primary separation mechanism near the wall is primarily driven by the gravitational and centrifugal force-induced momentum. However, surging causes a breakdown in this mechanism by swapping this primary flow and separation behaviour, where the pressure drop becomes the primary driver of the flow near the walls and gravitational and centrifugal force-induced momentum primarily drives the flow in the core of the hydrocyclone.

**Keywords:** hydrocyclone; surging; computational fluid dynamics; granular dynamics; air-core; discrete element method

## 1. Introduction

Hydrocyclones, gas–solid cyclones, and dense medium cyclones (DMCs) are used extensively in industrial separation applications such as waste-water treatment, mineral separation, and in the petrochemical industry, amongst others [1–4]. Consequently, much research attention has been given to studying the complex multiphase interactions involved in the separation process of hydrocyclones [1–3,5–7] as well as the optimisation of gas–solid, dense medium, and hydrocyclone design [4,8–14].

Cyclone separator flow fields are characterised by highly swirling turbulent flows and consequent flow separation with resulting secondary flows [1,5,15]. This turbulent spiral-swirling flow [16] leads to a primary stream that flows from the inlet to the spigot/underflow, along the separator wall [16] and separates into a secondary flow stream, which is further fed by short-cut flow that separates from the inlet stream, flowing to the centre of the cyclone and through to the vortex finder and overflow [16].

In addition to the complex swirling turbulent flow behaviour, multiphase interactions play a critical role in the behaviour of cyclone separators [1–3,5–7,15]. The multiphase interactions comprise particle–fluid, inter-particle, and particle–wall interactions [5–7,14,15,17]. Entrained in the feed flow, the particles separate along with the primary and secondary streams [16], whereby the fine particles primarily entrain in the secondary stream and the coarse particles primarily entrain in the primary stream [16,18]. In the case of hydrocyclones and DMCs, an additional phenomenon known as air-core formation [16,17] occurs in the core of the separator with resulting liquid–gas interactions which occur at the air–water interface [5,15,17].

Air-core formation occurs predominantly due to the pressure drop between the inlet and the outlets (overflow and underflow) [17]. Coupled with the turbulent spiral-swirling flow, a low pressure region forms in the core of the hydrocyclone/DMC. This region occurs along the central axis [16,17] of the separator resulting in air ingress from the underflow and the overflow which combined with air entrained in the feed of the hydrocyclone forms an upward-rotating column of air, known as the air-core, in the separator core [16,17]. Despite being an unsteady and often unstable flow phenomenon [17], the air-core stabilizes the vortex flow [16] and aids in the separation mechanisms.

Computational methods have been widely used to study the complex phenomena in cyclone separators. Specifically, computational fluid dynamics (CFD) has been used extensively to study the fluid dynamics in gas–solid cyclones, hydrocyclones, DMCs [1–3,5–7,19,20], and the associated liquid–gas interactions due to air-core formation [1,2,17]. The most common approach used for modelling the air-core formation is the Volume-Of-Fluid (VOF) model [2,5,6,17,21,22]. The inter-particle, particle–wall, and particle–fluid interactions have also been widely studied by coupling CFD models with appropriate computational granular dynamic (CGD) models, most commonly coupling CFD with the discrete element method (DEM) [1,5–7,15,19,20,23].

In the context of hydrocyclone and DMCs, much attention has been given in the literature on modelling the above complex fluid dynamics and multiphase interactions. Chu et al. [20] focused on comparing the predictions of the VOF-DEM and Mixture-DEM model (with a coarse-grained particle approach) for modelling relatively dilute (in terms of percentage solids) and dense flows, to see which model is best suited to the different scenarios in a hydrocyclone. However, Chu et al. [20] did not specifically study the phenomenon of surging and how surging modifies the dynamic behaviour of the cyclonic flow in a hydrocyclone. Thus, the phenomenon of surging has not received much treatment in the literature, particularly the effect of surging on the modification of the dynamic behaviour of the cyclonic flow in a hydrocyclone and/or DMC, more specifically, the effect that surging has on the air-core and the resulting separation mechanisms. The phenomenon of surging in a hydrocyclone or DMC is a direct consequence of the accumulation of particles, and subsequent increase in mass loading, in the cone and spigot of the hydrocyclone or DMC [23]. In many hydrocyclone applications, the intention is to operate the hydrocyclone at the highest particle feed rate without surging. Thus, modelling surging may provide insight into the effect of such on hydrocyclone performance. This could lead to future work to improve hydrocyclone design and operating conditions. Therefore, the aim in this study is to determine how the surging modifies the dynamic behaviour of the cyclonic flow in a hydrocyclone using coupled CFD and CGD models, specifically a two-way coupled VOF-DEM model. For turbulence closure, the Reynold's Stress Model (RSM) and the Large Eddy Simulation (LES) were used. The model predictions were benchmarked against each other in relation to and validated against experimental measurements.

## 2. Materials and Methods

### 2.1. Experimental Setup and Results

The primary purpose of the experimental work was to provide results to benchmark the simulation results against and to validate the simulations. The experiments were conducted on a 100 mm (barrel) diameter hydrocyclone as shown in Figure 1. The experiments conducted were standard gravimetric tests where the feed pressure and solid concentration were kept constant and the feed, overflow, and underflow mass flow rates were measured.

The feed pressure was measured via the analogue feed pressure gauge. The feed valve was opened steadily until the required feed pressure was obtained and maintained at 150 kPa. The solid density was 3330 kg/m$^3$ and the target solid loading (by mass) at the feed was 25%. The tests were repeated under the same conditions to obtain statistically representative and repeatable measurements as well as to quantify the experimental error. The corresponding feed, underflow, and overflow mass flow rates and mass fractions are given in Table 1.

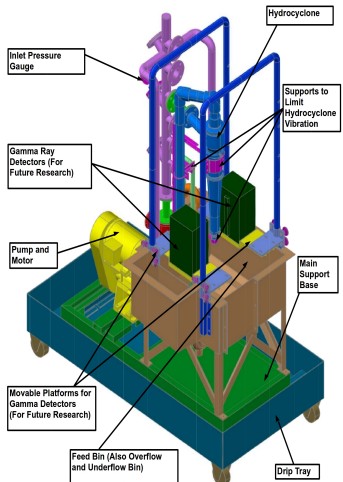

**Figure 1.** Experimental setup (adapted from an image courtesy of Multotec Pty Ltd., Johannesburg, South Africa) [24].

**Table 1.** Feed, overflow, and underflow mass flow rates as well as feed mass fractions.

| Measurement | Value |
|---|---|
| Feed Mass Flow Rate Water (kg/s) | $5.23 \pm 0.54$ |
| Feed Mass Flow Rate Solids (kg/s) | $1.65 \pm 0.11$ |
| Feed Water Mass Fraction (%) | $76.05 \pm 2.01$ |
| Feed Solids Mass Fraction (%) | $23.95 \pm 2.01$ |
| Overflow Mass Flow Rate Water (kg/s) | $4.38^{+1.01+0.54}_{-0.49-0.54}$ |
| Overflow Mass Flow Rate Solids (kg/s) | $0.14^{+0.03+0.02}_{-0.02-0.02}$ |
| Underflow Mass Flow Rate Water (kg/s) | $0.85^{+0.09+0.06}_{-0.11-0.06}$ |
| Underflow Mass Flow Rate Solids (kg/s) | $1.51^{+0.16+0.11}_{-0.19-0.11}$ |

Due to the nature of gravimetric tests, it is difficult to characterise the true experimental error. Thus, one way to characterise a form of experimental error is to run the test at the same operating conditions repeatedly to determine a mean value and statistically determine the upper and lower limits of the measured value and the standard deviation. Thus, the values in Table 1 present the mean values plus or minus values that represent the upper and lower limit of the experimentally measured values which are determined from the minimum and maximum measured values as well as the standard deviation.

Samples from the feed were collected for particle size analysis. The particle size analysis was conducted on a Mastersizer Hydro 2000 G at Multotec Pty Ltd. The resulting particle size distribution (PSD) at the feed is given in Table 2 and shown in Figure 2.

**Table 2.** Feed particle size distribution—cumulative passing.

| | | | | | | Particles Sizes in μm | | | | | | | |
|---|---|---|---|---|---|---|---|---|---|---|---|---|---|
| | **2** | **10** | **16** | **38** | **53** | **75** | **106** | **150** | **212** | **300** | **425** | **600** | **850** |
| **Sample** | | | | | | **Percentage Mass Below Sizes—Accuracy $\pm$ 1%** | | | | | | | |
| Test 1 | 2.65 | 7.32 | 8.89 | 13.44 | 17.91 | 25.75 | 37.10 | 51.53 | 67.66 | 83.03 | 94.34 | 99.63 | 100 |
| Test 2 | 0.54 | 3.64 | 4.91 | 8.37 | 11.20 | 15.97 | 23.43 | 35.16 | 52.29 | 72.79 | 90.19 | 99.01 | 100 |
| Test 3 | 0.00 | 1.00 | 1.70 | 4.92 | 8.20 | 13.93 | 22.69 | 35.52 | 53.03 | 73.23 | 90.21 | 98.94 | 100 |
| Average | 1.07 | 3.99 | 5.17 | 8.91 | 12.44 | 18.55 | 27.74 | 40.74 | 57.66 | 76.35 | 91.58 | 99.19 | 100 |

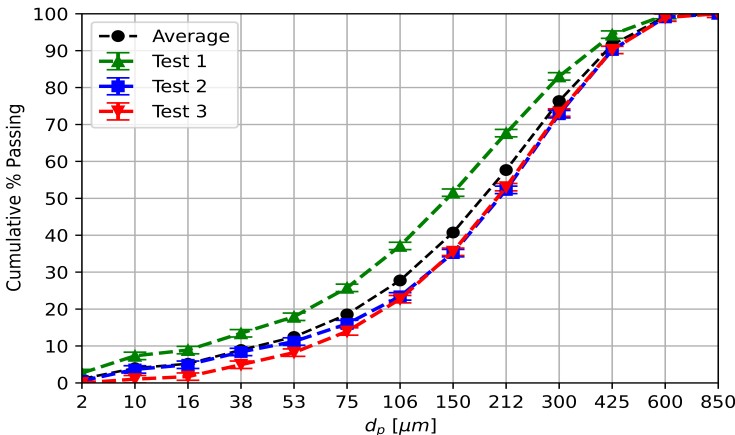

**Figure 2.** Feed particle size distribution—cumulative passing.

## 2.2. Model Geometry and Mesh

The model geometry, salient boundary locations, and the computational mesh are shown in Figure 3. A 100 mm barrel diameter hydrocyclone was modelled. A full hexahedral mesh is used in all regions whereby a mixture of structured and unstructured quadrilaterals are used for the face meshes. The initial cell size was 5 mm and the mesh was successively refined (3.75 mm and 2.5 mm) for the mesh sensitivity study. Mesh independence was achieved on the 2.5 mm mesh. Whilst this is arguably a coarse-grained mesh, Chu, Chen and Yu [7] and Chu et al. [25] demonstrated that such approaches yield qualitatively meaningful results and in the validation discussion of this study, it will be demonstrated that quantitatively valid results can be obtained on such a coarse-grained mesh. This is beneficial as it alleviates the concern of computational cost on fine meshes in such highly coupled multiphase models.

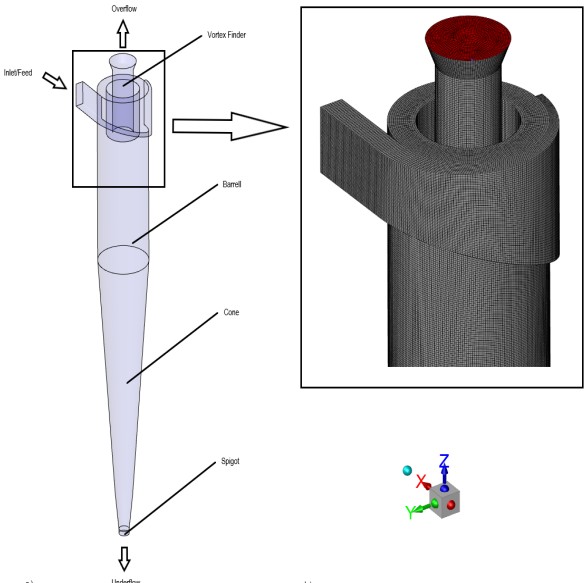

**Figure 3.** Hydrocyclone (**a**) geometry and (**b**) computational mesh.

## 2.3. Governing Equations

### 2.3.1. Conservation of Mass, Conservation of Momentum, and Turbulence

The Reynold's Averaged Navier–Stokes (RANS) Equations (1) and (2) describe the conservation of mass and momentum for a turbulent incompressible flow, respectively [26–31]. The instantaneous velocities $u_i$ are a superposition of the mean ($\bar{u}_i$) and fluctuating velocity

components ($u_i'$) as shown in Equation (3) [26–30]. Turbulence is accounted for in this study using two classes of models, namely the RANS and LES approaches.

$$\frac{\partial \rho}{\partial t} + \nabla.(\rho \bar{\mathbf{u}}) = 0 \tag{1}$$

$$\rho \frac{\partial \bar{u}_i}{\partial t} + \rho \nabla.(\bar{u}_i \bar{\mathbf{u}}) = -\frac{\partial \bar{p}}{\partial x_i} + \nabla.(\mu \nabla \bar{u}_i) - B_i - \rho \frac{\partial}{\partial x_i}(\overline{u_i' u_j'}) + S_M \tag{2}$$

for $i, j, l = x, y, z$.

$$u_i = \bar{u}_i + u_i' \tag{3}$$

Closure to the RANS equations was modelled with the RSM which is a seven-equation model for three-dimensional systems [29,32,33]. The RSM constitutes the transport equations for the turbulent kinetic energy $k$ (Equation (4)) and the six Reynolds stresses $-\rho \overline{u_i' u_j'}$ (Equation (5)) [29,32,33]:

$$\rho \frac{\partial k}{\partial t} + \rho \nabla.(k\mathbf{u}) = \nabla.\left(\mu + \frac{\mu_t}{\sigma_k}\nabla k\right) + \frac{1}{2}(P_{ii} + G_{ii}) - \rho \epsilon (1 + 2M_t^2) + S_k \tag{4}$$

$$\rho \frac{\partial \overline{u_i' u_j'}}{\partial t} + C_{ij} = D_{T,ij} + D_{L,ij} - P_{ij} + \phi_{ij} - \epsilon_{ij} - F_{ij} + S_{\rho \overline{u_i' u_j'}} \tag{5}$$

The Stress-Omega pressure strain term ($P_{ij}$) was employed due to its proven accuracy for highly swirling flows and flows over curved boundaries [29].

For the LES approach, the static Smagorinsky LES model was used. The LES model uses a space (Favré)-filtered form of the Navier–Stokes equations as given by Equations (6) and (7) [29,32,33]:

$$\frac{\partial \rho}{\partial t} + \nabla.(\rho \bar{\mathbf{u}}) = 0 \tag{6}$$

$$\rho \frac{\partial \bar{u}_i}{\partial t} + \rho \nabla.(\bar{u}_i \bar{\mathbf{u}}) = -\frac{\partial \bar{p}}{\partial x_i} + \frac{\partial \tau_{ij}}{\partial x_j} + \frac{\partial \sigma_{ij}}{\partial x_j} - B_i + S_M \tag{7}$$

The stress tensor, due to molecular viscosity, is given by [29]:

$$\sigma_{ij} \equiv \left[\mu\left(\frac{\partial \bar{u}_i}{\partial \bar{x}_j} + \frac{\partial \bar{u}_j}{\partial \bar{x}_i}\right)\right] - \frac{2}{3}\mu \frac{\partial \bar{u}_l}{\partial \bar{x}_l}\delta_{ij} \tag{8}$$

and the filtered *SGS* stress tensor is given by [29,34]:

$$\tau_{ij} \equiv \rho \overline{u_i u_j} - \rho \bar{u}_i \bar{u}_j \tag{9}$$

The filtered *SGS* is computed using the Boussinesq hypothesis [29,34]:

$$\tau_{ij} - \frac{1}{3}\tau_{kk}\delta_{ij} = -2\nu_t \mathbf{S}_{ij} \tag{10}$$

Using the Static Smagorinsky–Lilly Model, the eddy-viscosity is calculated using [29]:

$$\nu_t = \rho L_s^2 |\mathbf{S}| \tag{11}$$

where the magnitude of the rate-of-strain tensor is given by $|\mathbf{S}| \equiv \sqrt{2\mathbf{S}_{ij}\mathbf{S}_{ij}}$ and the mixing length $L_s = min(\kappa d_w, C_s V_c^{1/3})$ [29]. The Smagorinsky constant is $C_s = 0.14$. Only gravity was accounted for in the body force terms; thus, $B_x = B_y = 0$ and $B_z = -\rho \mathbf{g}$.

2.3.2. Volume-of-Fluid Model for Air-Core Formation

In this study, all the models were setup with the multiphase model for the the air-core from the outset as opposed to the two-step process commonly used in the literature such as in [2,5,15,17,21,22]. The models were initialised under the assumption of a hydrocyclone filled with air prior to operation. Thus, the entire model was initialised with $\alpha_{air} = 1$. The model was then run till steady state. This produced the realistic transient phenomena of air-core formation. This single step process ensured that the air-core formed in every model without the divergent behaviour observed when using the two-step process as used in [2,5,15,17,21,22].

The different fluid phases, in the case of this study—water and air, are treated as non-interpenetrating continua in the VOF models wherein only the interface between the phases is tracked [29]. Thus, the continuity and momentum equations are solved for the water-air mixture as opposed to each phase [29]. The mixture flow field is described by Navier–Stokes equations (Equations (1) and (2) for RANS and Equations (6) and (7) for LES). The phases are decoupled via an additional transport equation for the volume fraction for the air phase:

$$\frac{\partial}{\partial t}(\alpha_{air}\rho_{air}) + \nabla.(\alpha_{air}\rho_{air}\mathbf{u}_{air}) = 0 \tag{12}$$

It follows that the volume fraction for the water phase is determined from $\alpha_{air} + \alpha_w = 1$ [29]. The Continuum Surface Stress (CSS) model was used to account for surface tension as in [35]. Under the assumption of constant surface tension coefficient, as in the model used in this study, the surface tension force is given by Equation (13):

$$F_{CSS} = \sigma\nabla.\left(|\nabla\alpha|I - \frac{\nabla\alpha \otimes \nabla\alpha}{|\nabla\alpha|}\right) \tag{13}$$

The surface tension force ($F_{CSS}$) is added to the momentum equation (Equations (2) or (7)) via the momentum source term ($S_M$). The volume fraction ($\alpha$) is either the volume fraction of water or air, depending on which phase $F_{CSS}$ is being computed for. The surface tension coefficient was set at $\sigma = 0.073$ N/m. Wall adhesion was added to the model The contact angle depends on the solid interface properties. In this case, the material is polyurethane (PU). According to [36], the contact angle for PU can range from 71.1° to 94.2°, but it depends on various factors including surface treatment. Thus, without further knowledge of the factors that impact the contact angle, the contact angle can only be assumed as some value within the above range. The contact angle was set as a default 90°. Given that the contact angle only comes into play when the water–air interface occurs at a solid surface; in the context of the hydrocyclone, this is likely to occur only near the underflow and overflow, thus, arguably having minimal broader effect. The contact angle is used to modify the surface normal in near wall cells, thus, adjusting the near-wall surface curvature [29]. The adjusted local curvature adjusts the body force term when calculating the surface tension [29]. The surface normal is adjusted as per Equation (14) [29]:

$$\hat{n} = \hat{n}_w cos\theta_w + \hat{t}_w sin\theta_w = \hat{n}_w cos(90°) + \hat{t}_w sin(90°) = \hat{t}_w, \tag{14}$$

where $\bar{n} = \nabla\alpha$ and $\hat{n} = \bar{n}/|\bar{n}|$ [29].

2.3.3. Discrete Element Method for Particle Interactions

The particles are modelled using the discrete phase model (DPM) with discrete element method (DEM) collision rules. The particles are modelled as spherical particles dispersed in the continuous phase(s) [29]. Newton's equations of motion are applied to each particle and the particles' velocity fields and the trajectories are obtained from the integration of the equations of motion for the particles [29].

The particle(s) equation of motion, which is an ordinary differential equation (ODE) given in Equation (15) accounts for drag, gravity, virtual mass effects, and pressure gradient

forces which account for the momentum transfer from the fluid to the particles (fluid–particle interaction) [29]. Similarly, the momentum transfer from the particles to the fluid is accounted for in the momentum source term $S_M$ in Equations (2) and (7) [29], thus, resulting in a two-way coupled CFD-CGD model.

$$m_p \frac{d\mathbf{u}_p}{dt} = m_p \frac{18\mu}{\rho_p d_p^2} \frac{C_D Re}{24} (\mathbf{u} - \mathbf{u}_p) + m_p \frac{\mathbf{g}(\rho_p - \rho)}{\rho_p} + m_p C_{vm} \frac{\rho}{\rho_p} \left( \mathbf{u}_p \nabla \mathbf{u} - \frac{d\mathbf{u}_p}{dt} \right) + m_p \frac{\rho}{\rho_p} \mathbf{u}_p \nabla \mathbf{u} + \bar{F}_{other} \quad (15)$$

The drag coefficient $C_D$ was calculated using the spherical drag law as defined in [29,37] and is given by the following equation:

$$C_D = \sum_{i=1}^{3} \frac{a_i}{Re^{i-1}} \quad (16)$$

where the coefficients $a_i$ are given in [37] and are determined based on the local fluid Reynold's Number.

Turbulence has an effect on the dispersion of the particles [6,29] and this was incorporated in the model via the stochastic tracking or discrete random walk model as used in [6] and detailed in [29]. By doing so, not only is the effect of the mean fluid velocity ($\bar{u}_i$) on the particle accounted for, but the effect of the turbulent fluctuating component of the fluid velocity ($u_i'$) on the particle is accounted for as well (effectively using the instantaneous velocity and not just the mean velocity of the fluid) [6,29].

Thus, the fluctuating component is calculated by sampling over the turbulent eddy lifetime and assuming that these velocities obey a Gaussian probability distribution ($\zeta$ is a normally distributed random number) [6,29]:

$$u_i' = \zeta \sqrt{\overline{u_i'^2}} \quad (17)$$

The fluctuating components presented in Equation (17) are applicable for the RSM [29] where nonisotropy in the turbulent stresses are accounted for [29]. However, for the LES, the velocity fluctuations are considered isotropic [29].

The source term $\bar{F}_{other}$ from Equation (15) accounts for the forces due to inter-particle and particle–wall collisions [29] and are incorporated in the DEM. The normal forces arising from the collisions are modelled as a spring–dashpot system and the tangential forces are modelled via a friction collision law [29]. The normal forces from a collision between arbitrary particle $i$ and $j$ are given Equations (18) and (19) [29,38]:

$$\bar{F}_{ij} = (K\delta + \gamma(\bar{v}_{ij} \cdot \bar{e}_{ij}))\bar{e}_{ij} \quad (18)$$

$$\bar{F}_{ij} = -\bar{F}_{ji} \quad (19)$$

The tangential (friction) force is calculated from Equation (20) [29]:

$$\bar{F}_{friction} = \mu_f \bar{F}_{ij} \quad (20)$$

where $\mu$ is the friction coefficient between the two colliding particles. The particle–wall collisions are modelled in the same way as particle–particle collisions except that the wall has different material properties from the particle.

The Ansys particle-parcel method [39] was used which, as stated in [20], is similar in principle to the coarse-grained treatment. Thus, the individual particles were not modelled, rather 'parcels' of particles were modelled. In this approach, a 'parcel' of particles is modelled as opposed to modelling each particle based on the mass flow rate [39]. Thus, the 'parcel' represents a certain fraction of the mass flow rate of particles [39]. Based on this approach, the number of parcels in the model ranged from 66, 493 at a flow time of 0.1 s to 782,118 at a flow time of 4 s for the VOF-DEM RSM model.

### 2.3.4. Boundary Conditions

The boundary conditions are given in Table 3. Atmospheric Pressure $(P_a)$ is taken as $P_a = 82.5$ kPa which was measured using a barometer. The lower atmospheric pressure is due to the high altitude at which the test was conducted. The inlet conditions were specified based on the experimental measurements.

**Table 3.** Flow boundary conditions.

| Boundary Name | Boundary Type | Pressure (kPa) | $\alpha_{air}$ | $\dot{m}$ (kg/s) | $D_H$ (m) | $I_{turb}$ (%) |
|---|---|---|---|---|---|---|
| Inlet | Mass-flow inlet | 150 | 0 | 6.875 ($\dot{m}_w = 5.23$) | 0.02 | 8.86 |
| Underflow | Pressure outlet | $P_a$ | 1 (backflow) | N/A | 0.02 | 10 |
| Overflow | Pressure outlet | $P_a$ | 1 (backflow) | N/A | 0.075 | 10 |

The inlet mass flow rate is a superposition of the water and particle mass flow rates at the inlet and the water–particle mass split at the feed is based on the mass fractions given in Table 1. A surface injection was used at the inlet to introduce the particles. The inlet PSD was specified by fitting the experimentally measured PSD to the Rosin–Rammler distribution given in Equation (21) and by applying the fitted distribution to the inlet boundary for the particles.

$$Y_d = e^{(-d/\bar{d})^n} \tag{21}$$

where $Y_d$ is the mass fraction of particles with diameter greater than $d$, and $\bar{d}$ is the mean diameter [39]. In this case, the mean diameter is the diameter at which $Y_d = e^{-1} \approx 0.368$. The spread parameter $n$ is determined by Equation (22) [39].

$$n = \frac{\ln(-\ln(Y_d))}{\ln(d/\bar{d})} \tag{22}$$

As per the procedure outlined in the ANSYS Fluent User Guide [39], the spread parameter $n$ is calculated for each size bin and then averaged. The average $n$ is used in the Rosin–Rammler distribution. Fitting the average data from Table 2 produced the Rosin–Rammler distribution parameters given in Table 4. The average experimental PSD and the fitted distribution are shown in Figure 4.

**Table 4.** Rosin–Rammler Distribution Parameters.

| Boundary | $\bar{d}$ (μm) | $n_{ave}$ | $d_{min}$ (μm) | $d_{max}$ (μm) | $N$ |
|---|---|---|---|---|---|
| Feed | 240 | 1.3285 | 2 | 850 | 13 |

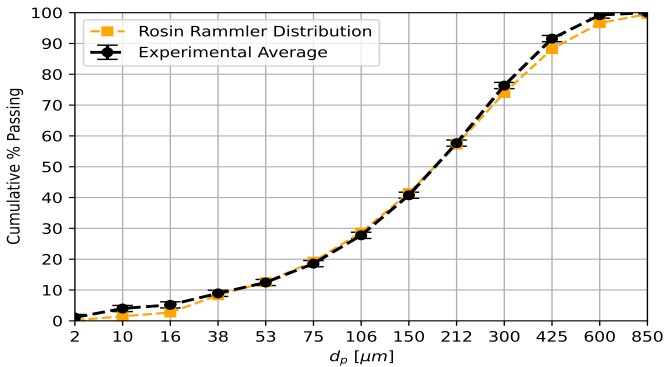

**Figure 4.** Feed PSD—experimental average against the Rosin–Rammler distribution fit.

At the outlets, a radial equilibrium pressure distribution is applied. The implication of this is that the gauge pressure is applied at the boundary centre and the static pressure elsewhere on the boundary is calculated using [39]:

$$\frac{\partial p}{\partial r} = \frac{\rho v_\theta^2}{r}$$

(23)

The no-slip condition was specified, for the fluid, at the wall boundaries, therefore:

$$\mathbf{u}_{wall} = \mathbf{0}$$

(24)

### 2.4. Solver Setup

The Quadratic Upstream Interpolation for Convective Kinematics (QUICK) discretisation scheme was used for the momentum, energy, turbulent kinetic energy, and the Reynold stress equations. Pressure correction was done using the PRESTO! discretisation scheme. The Geo-Reconstruct scheme was used for the VOF interface reconstruction. The least squares cell-based method was used for Gradient reconstruction. The implicit body force treatment was used for the VOF body force formulation. The default under-relaxation factors were used as per [39].

To ensure that convergence was achieved at each time step within 5–15 iterations, the time step size of $(\Delta t = 5 \times 10^{-5}\text{s})$ was used. The transient formulation was bounded second-order implicit. The particles were introduced using a particle time step that was different from the fluid time step size, specifically 100 times the fluid time step size, to make the model computationally tractable. However, the two–way coupled fluid–particle momentum transfer was updated at every flow iteration.

## 3. Results and Discussion

### 3.1. Model Validation

Table 5 presents the average mass flow rate of water at the underflow and overflow, as measured from experiment, as predicted by the models and the associated error between the model predictions in relation to experiment. Figures 5 and 6 present the time evolution of the water mass flow rate predictions at the overflow and underflow, respectively. The results demonstrate that the model predictions are quantitatively valid and within bounds of the experiment for the overflow and underflow water mass flow rate. However, the LES model does exhibit noticeable divergence from the experiment at the underflow and spikes in both the overflow and underflow water mass flow rates which are well below the minimum or well above the maximum values measured experimentally. Thus, the results in Table 5 as well as Figures 5 and 6 indicate that the VOF-DEM model, with the RSM, provides the most accurate predictions in relation to the experiment in terms of the water mass flow rate at the outlets.

**Table 5.** VOF-DEM water mass flow rate model comparisons.

| Model | $m_{under}$ (kg/s) | $m_{over}$ (kg/s) | $m_{under,exp}$ (kg/s) | $m_{over,exp}$ (kg/s) | $\|Error_{under}\|$ (%) | $\|Error_{over}\|$ (%) |
|---|---|---|---|---|---|---|
| VOF-DEM (LES) | 0.41 | 4.78 | $0.85^{+0.09+0.06}_{-0.11-0.06}$ | $4.38^{+1.01+0.54}_{-0.49-0.54}$ | 49 | 10 |
| VOF-DEM (RSM) | 0.79 | 4.44 | $0.85^{+0.09+0.06}_{-0.11-0.06}$ | $4.38^{+1.01+0.54}_{-0.49-0.54}$ | 7 | 2 |

The above result may appear counter-intuitive as it would be expected that the LES results should be more accurate than the RSM results, particularly in a complex turbulent flow such as in a hydrocyclone because LES models resolve a larger range of turbulent eddies, whereas the RSM models a larger range of turbulent eddies. However, the results in the above models are on meshes that are still too coarse for quantitatively accurate predictions from LES models. Despite this, the LES models provide meaningful qualitative predictions which are explored and demonstrated in the next section.

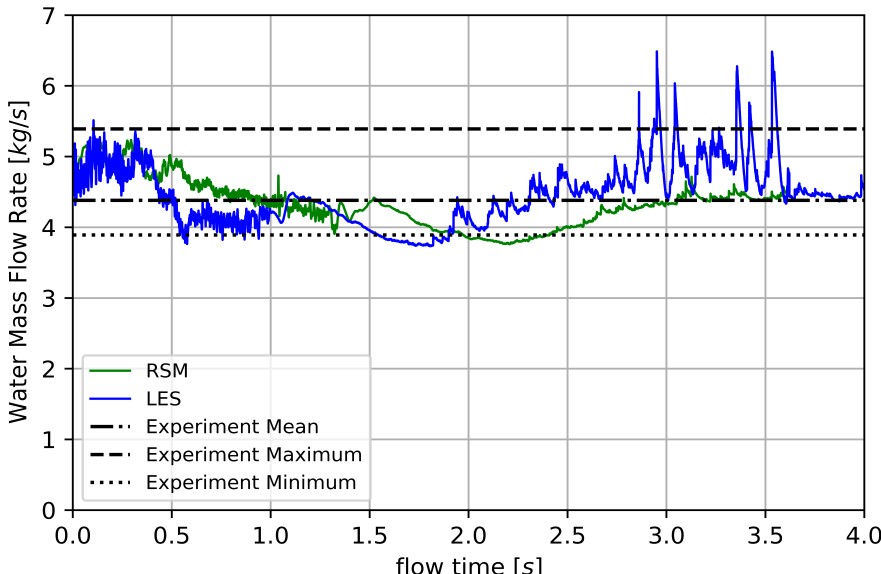

**Figure 5.** Overflow water mass flow rate as predicted by VOF-DEM models in relation to experiment.

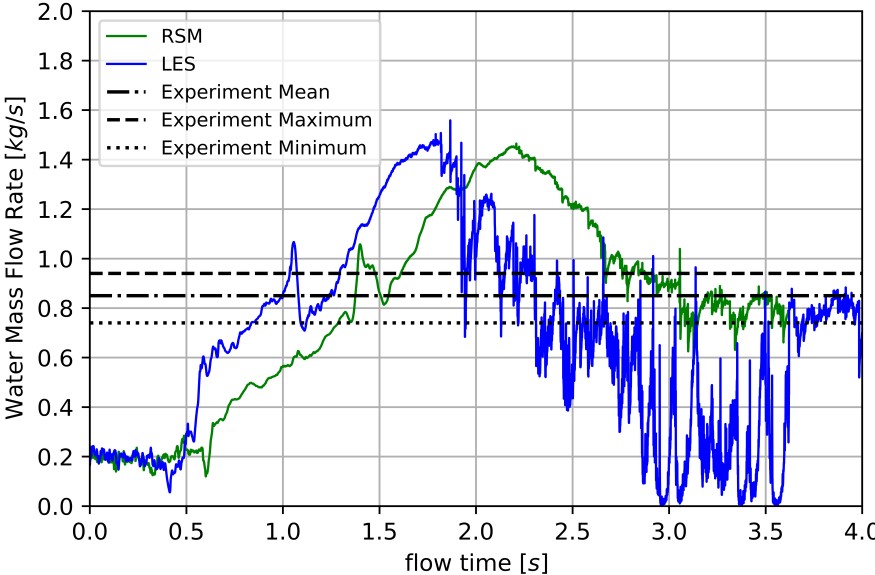

**Figure 6.** Underflow water mass flow rate as predicted by VOF-DEM models in relation to experiment.

However, the good agreement between experiment and the models on a coarse grained mesh, when using the RSM, is consistent with and further validation of the results from [7,25]. Due to the excessive computing time required, LES-based results on meshes that are suitably fine were out of reach in this study and it will be reserved for future work. However, it is useful for practitioners, which need to produce CFD results on coarse meshes, to know that RSM-based models can produce results that are within acceptable limits and can be used to make both quantitatively and qualitatively valid predictions.

### 3.2. Modification of the Dynamics of the Cyclonic Flow Due to Surging

Figure 7 highlights the suppression of the air-core over time as particles enter the hydrocyclone. It can be seen that the air-core has completely dissipated 2 s after the particles enter the hydrocyclone. A negligible amount of air is entrained in the overflow 4 s after the particles enter the hydrocyclone as seen in the enlarged image of Figure 7e, which may indicate the potential reformation of the air-core. This could be a numerical

glitch, or it could be reversed flow at the overflow, which may or may not indicate the potential for air-core reformation. The results from the CFD solver indicated some amount of reverse flow at the overflow at this point in the simulation, thus ruling out a numerical glitch and supporting the premise of reverse flow at the overflow and potential air-core reformation. However, further study is needed to confirm air-core reformation and recovery from surging.

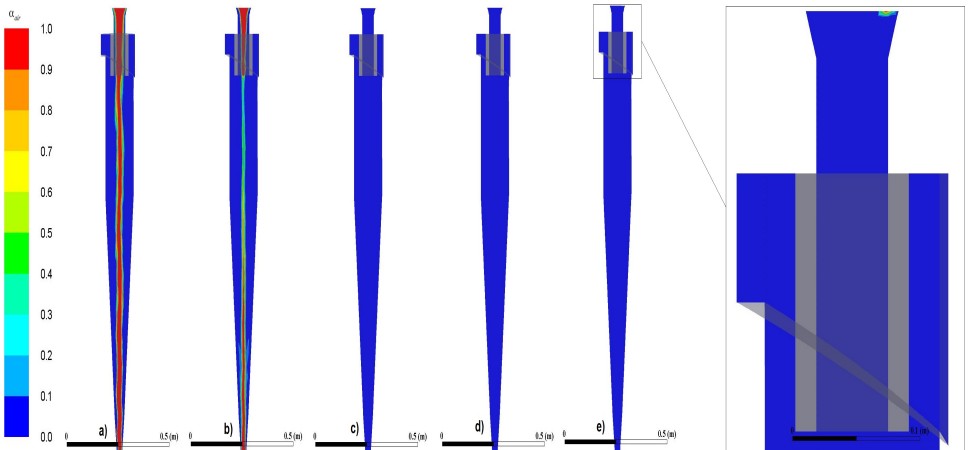

**Figure 7.** Contours coloured by volume fraction of air from the VOF-DEM model at flow-time after injection of particles: (**a**) 0 s, (**b**) 1 s, (**c**) 2 s, (**d**) 3 s and (**e**) 4 s.

This is a real phenomenon in hydrocyclones known as surging. In the case of surging, the particles accumulate in the hydrocyclone in the cone area specifically, as seen in the time evolving particle flow in Figure 8, where the particles in the cone region have a large residence time. This accumulation of particles causes the water velocity and the swirl in that region to decrease rapidly, resulting in a rise in pressure in the core of the hydrocyclone. The rise in pressure removes the low pressure region which is the primary mechanism for air-core formation. The concentration of particles and subsequent decrease in velocity and swirl and the rise in pressure can be seen in Figures 9–11. Whilst the VOF-DEM model predicts surging, it was not determined whether recovery from surging could be modelled.

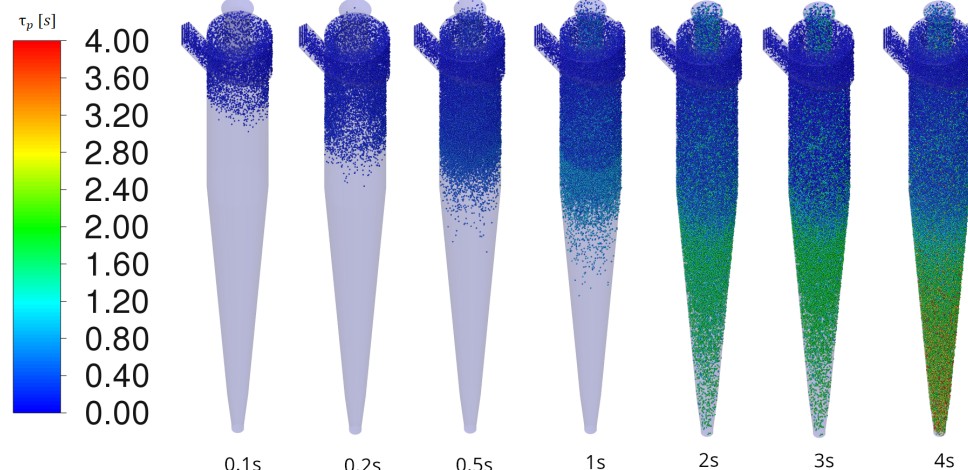

**Figure 8.** Particle flow coloured by particle residence time ($\tau_p$) at various flow-times after injection of particles.

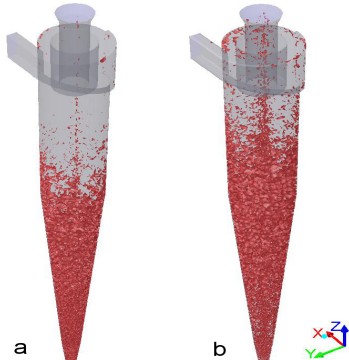

**Figure 9.** Iso-Surface of particle concentration above 600 kg/m³ from the VOF-DEM models at flow-time 4 s: (**a**) RSM and (**b**) LES.

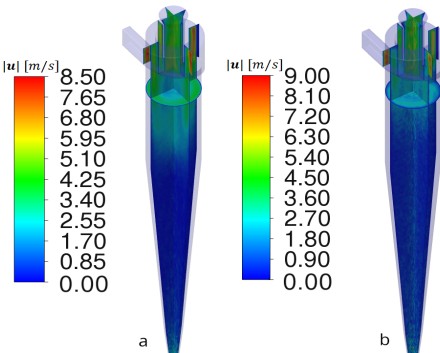

**Figure 10.** Contours coloured by velocity magnitude, on the planes $x = 0$, $y = 0$, and $z = 0$ from the VOF-DEM models at flow-time 4 s: (**a**) RSM and (**b**) LES.

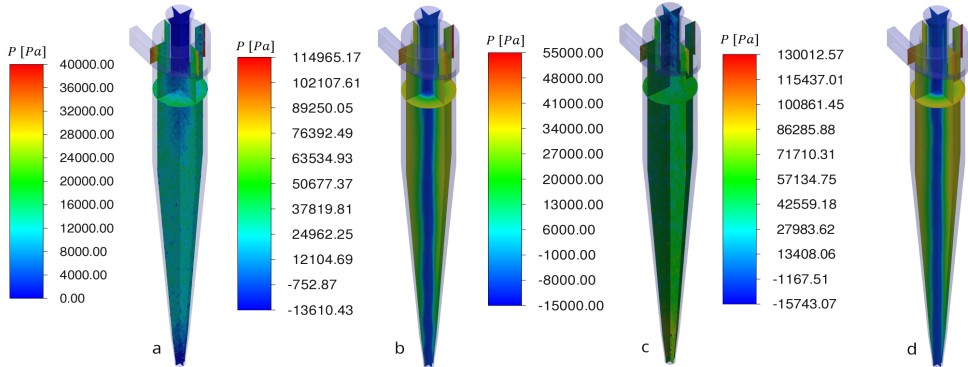

**Figure 11.** Contours coloured by static pressure, on the planes $x = 0$, $y = 0$, and $z = 0$ from the VOF-DEM models at flow-time 4 s and the VOF models at flow-time 2 s: (**a**) VOF-DEM RSM, (**b**) VOF RSM, (**c**) VOF-DEM LES, and (**d**) VOF LES.

Surging causes a significant change in the dynamic behaviour of the flow regime in the cyclone apart from the suppression of the air-core. Thus, the effect of the particles on the dynamic behaviour of the cyclonic flow was investigated by comparing the models with the particles against the corresponding air-core models. This was done in the case of the VOF and VOF-DEM models with both the RSM and LES turbulence models. The dynamic behaviour was investigated in terms of the tangential and axial velocities, the vorticity, Euler Number, and pressure. This was done by studying the pressure profiles through the core of the hydrocyclone and the velocities, vorticity, and Euler Number at four heights along the hydrocyclone, with varying $y/D$, namely $z/H = 0.2$, $z/H = 0.4$, $z/H = 0.6$, and $z/H = 0.8$ to correspond with positions near the underflow, higher up in the cone, in the barrel near the cone, and near the vortex finder, respectively. In addition, the pressure

profiles were taken as a function of height in the core of the hydrocyclone. The position of the places where the profiles were investigated is shown Figure 12. These profiles we investigated at flow-time 4 s for the simulations with the particles included and at flow-time 2 s for the air-core only simulations. For the air-core only simulations, the time stamp was chosen to correspond with when the air-core simulations reached steady state.

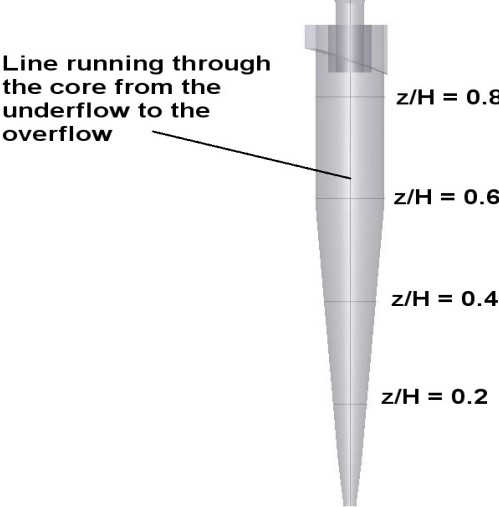

**Figure 12.** Hydrocyclone Geometry and positions where profiles were investigated.

Under particle–free air-core formation, the static and dynamic pressures are relatively constant along the height of the hydrocyclone. However, as seen in Figure 13, there is a noticeable drop in the static pressure and rise in the dynamic pressure near the vortex finder. The inclusion of particles, to the point of surging, causes a significant rise in the static pressure from the area near the vortex finder to the underflow and a significant rise in the dynamic pressure from the vortex finder to the overflow as seen in Figure 13. Apart from the apparent suppression of the air-core, this significant change in the pressure distribution leads to a significant impact on the velocity profiles and the vorticity in the hydrocyclone. The LES model displays excessive noise in the pressure distributions near the underflow during surging which may be part of the reason why the LES model predictions were less accurate than the RSM model in this study. However, this may be related to the relatively coarse mesh in the underflow region. It is expected that the LES model should provide more accurate results than the RSM model when the mesh is refined further.

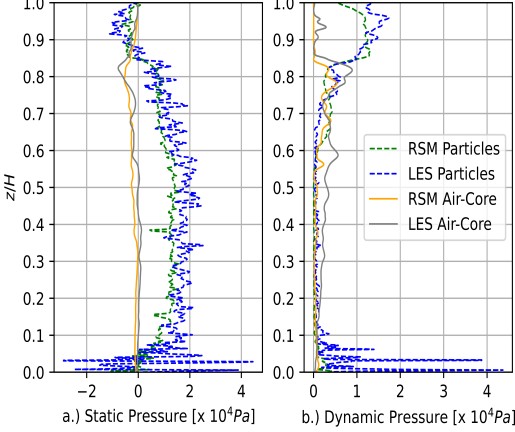

**Figure 13.** (**a**) Static and (**b**) dynamic pressure (Pa) as a function of the normalised $z$ co-ordinate at flow-time 4 s for the simulations with the particles included and at flow-time 2 s for the air-core only simulations.

The tangential and axial velocity profiles predicted by the VOF air-core models, as seen in Figures 14 and 15, match the expected profiles as reported in the literature. Under particle–free air-core formation, the peak tangential and axial velocities remain relatively constant at varying heights of the hydrocyclone. Figures 14 and 15 indicate that the RSM model predicts a peak tangential velocity that is up to 20% lower and a peak axial velocity that is up to 70% lower than the corresponding predictions by the LES model. Furthermore, the LES model more accurately represents the axial velocity profile near the underflow ($z/H = 0.2$).

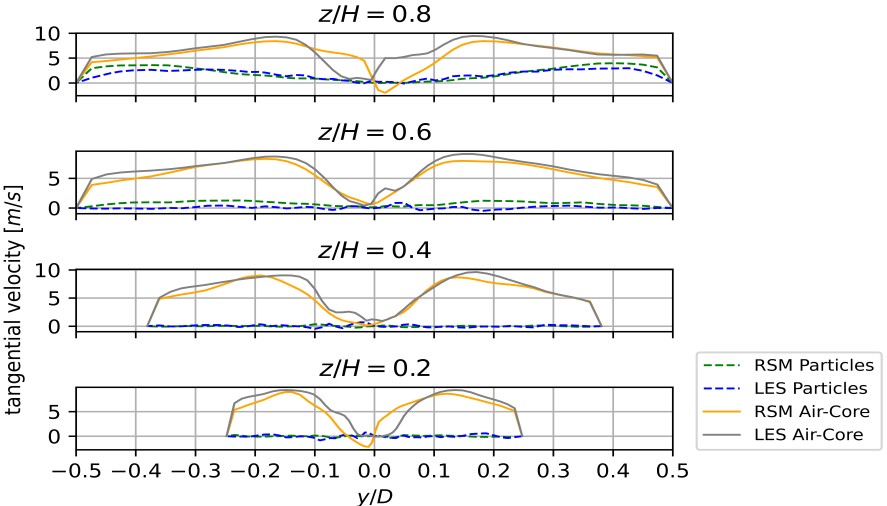

**Figure 14.** Tangential velocities (m/s) as a function of the normalised $y$ co-ordinate at various heights at flow-time 4 s for the simulations with the particles included and at flow-time 2 s for the air-core only simulations.

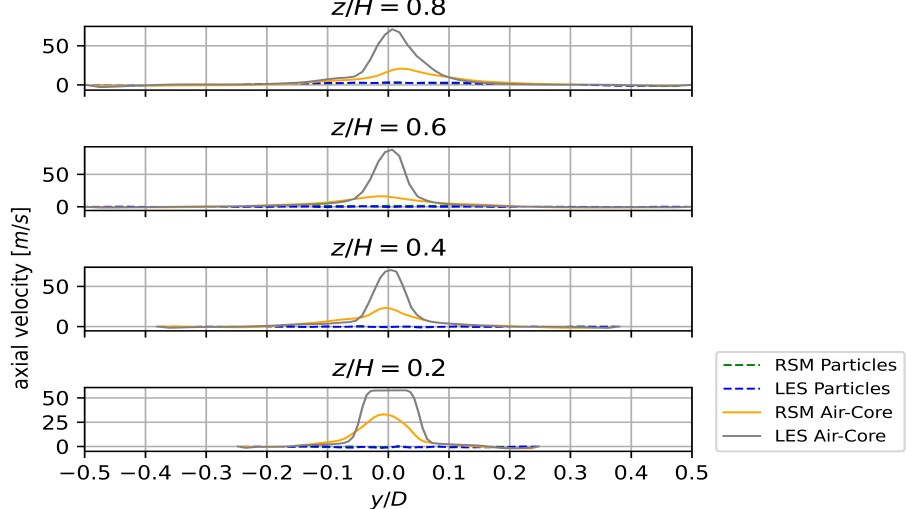

**Figure 15.** Axial velocities (m/s) as a function of the normalised $y$ co-ordinate at various heights at flow-time 4 s for the simulations with the particles included and at flow-time 2 s for the air-core only simulations.

When surging occurs, the tangential and axial velocities undergo significant attenuation. As seen in Figures 14 and 15, the only noticeable fluid motion occurs near the vortex finder ($z/H = 0.8$). This is once again due to the high concentration of particles in the rest of the hydrocyclone (as shown in Figures 9 and 10). Despite the attenuated velocity, the models still predict the 'M'-shaped tangential velocity profile as seen in Figure 14 at $z/H = 0.8$.

The subsequent attenuation of the vorticity, due to surging, can be seen in Figure 16. In the case of particle–free air-core formation, the vorticity peaks near the air–water interface due to the high level of separation in that region. The inclusion of the particles, to the point of surging, attenuates the vorticity in that region, subsequently breaking down the cyclonic separation. The LES model predicts a vorticity that is up to an order of magnitude higher than the RSM models.

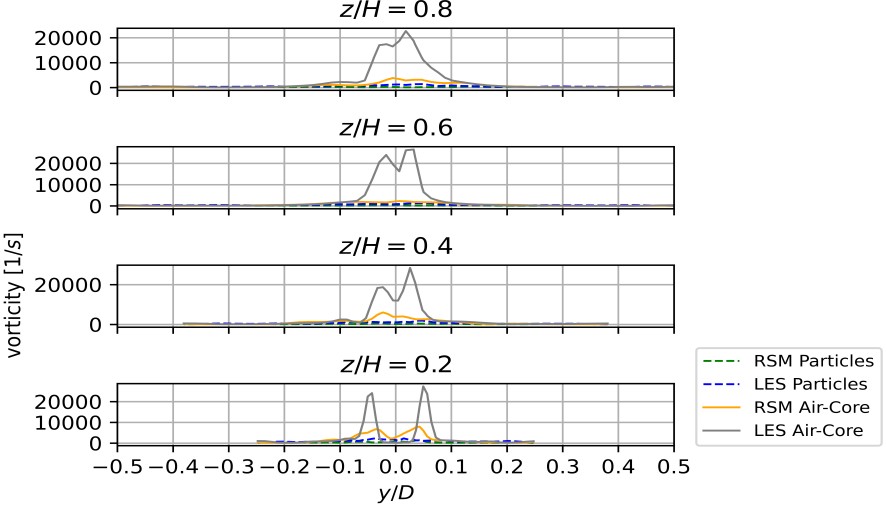

**Figure 16.** Vorticity magnitude (1/s) for the RSM and LES models as a function of the normalised $y$ co-ordinate at various heights at flow-time 4 s for the simulations with the particles included and at flow-time 2 s for the air-core only simulations.

The breakdown in the cyclonic separation can be described by the Euler Number. The variation of the Euler Number at various heights is given in Figures 17 and 18. The Euler Number was calculated as the ratio between the pressure drop from the inlet to the outlets and the dynamic pressure:

$$Eu = \frac{\Delta P}{(\rho u^2)/2},$$

(25)

where the pressure drop $\Delta P$ was calculated as the difference between the inlet pressure and the outlet pressure from Table 3 ($\Delta P = 150$ kPa $- P_a$) and the dynamic pressure was calculated by the CFD solver.

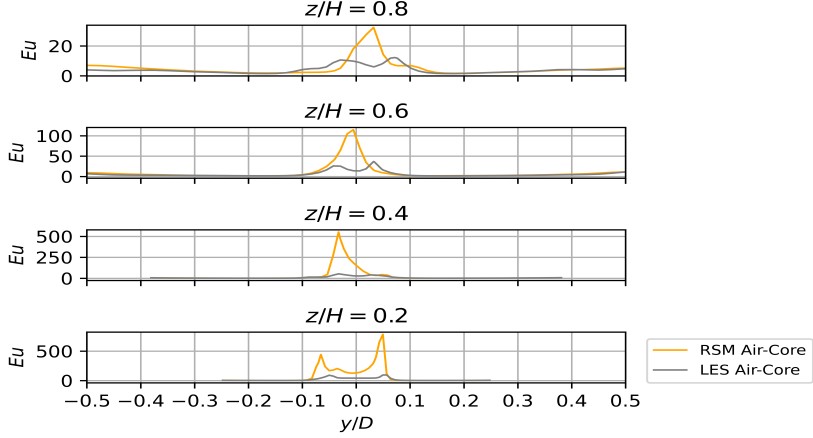

**Figure 17.** Euler number for the air-core models as a function of the normalised $y$ co-ordinate at various heights at flow-time 2 s.

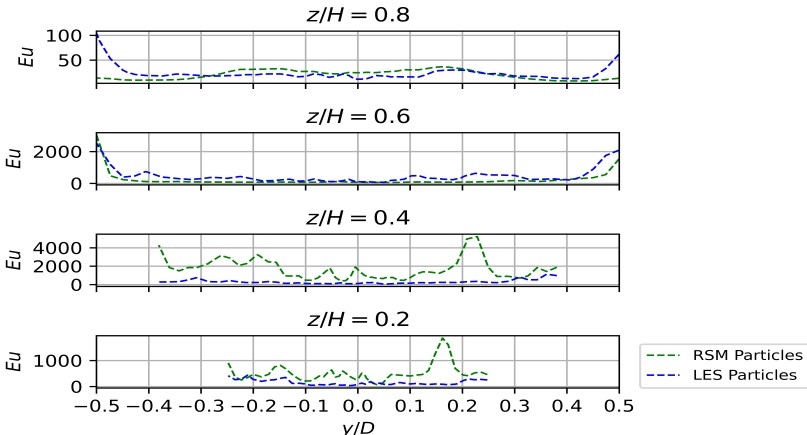

**Figure 18.** Euler number for the particle models as a function of the normalised $y$ co-ordinate at various heights at flow-time 4 s.

For both the LES and RSM models, the flow in the core of the hydrocyclone is driven primarily by the pressure drop as seen by the high Euler Number in the core of the hydrocyclone in Figure 17. Whereas the flow near the wall is primarily driven by the gravitational and centrifugal force-induced momentum as seen by the significantly lower Euler Number in the core of the hydrocyclone in Figure 17. This effect is somewhat reversed when the particles are added and surging occurs as seen in Figure 18. When surging occurs, the Euler Number rises near the walls and attenuates near the core, leading to a breakdown in the balance between the pressure and momentum-based separation in the core and near the wall of the hydrocyclone.

When investigating the dynamic behaviour of a hydrocyclone, it must be noted that the flow is not strictly symmetric about any plane or the azimuthal plane. Whilst there is some level of symmetry, strict symmetry cannot hold due to the position of the inlet, which causes a degree of asymmetry in both the geometry and boundary conditions. Consequently, symmetry cannot hold for both reality and in the simulations. However, there is similarity in the dynamic behaviour about various planes in terms of the flow as evidenced in the tangential velocities, axial velocities, the vorticity, and Euler Number about $y/D = 0$ as seen in Figures 14–18. Furthermore, when studying the profiles about another Azimuthal plane to the $yz$-plane, namely, the $xz$-plane, it can be seen that there is no strict symmetry in the flow fields but there is similarity in the dynamic behaviour of the flow. This can be seen when comparing the profiles on the $yz$-plane in Figures 14–18 to the corresponding profiles on the $xz$-plane in Figures 19–23.

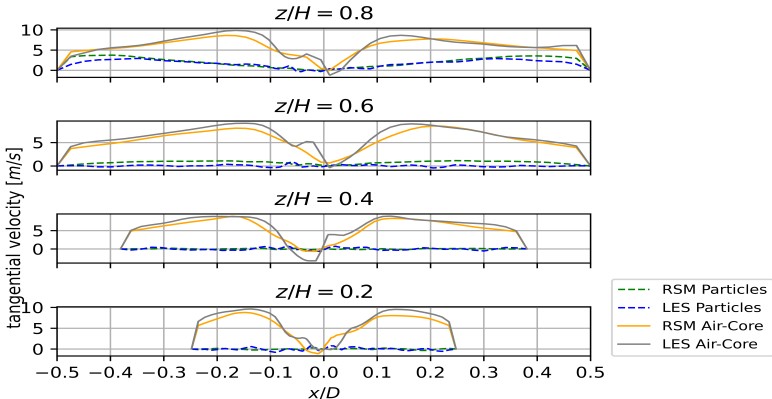

**Figure 19.** Tangential velocities (m/s) as a function of the normalised $x$ co-ordinate at various heights at flow-time 4 s for the simulations with the particles included and at flow-time 2 s for the air-core only simulations.

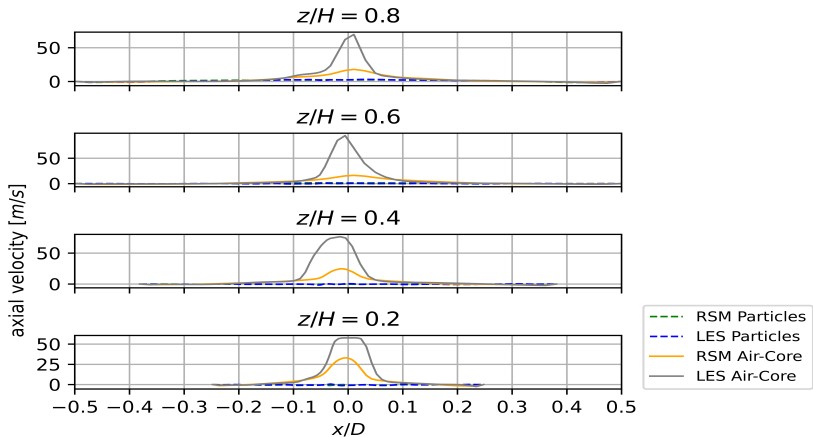

**Figure 20.** Axial velocities (m/s) as a function of the normalised $x$ co-ordinate at various heights at flow-time 4 s for the simulations with the particles included and at flow-time 2 s for the air-core only simulations.

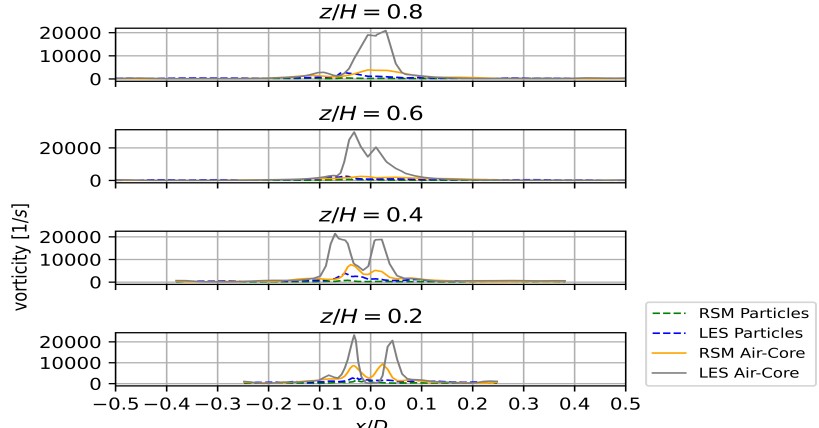

**Figure 21.** Vorticity magnitude (1/s) for the RSM and LES models as a function of the normalised $x$ co-ordinate at various heights at flow-time 4 s for the simulations with the particles included and at flow-time 2 s for the air-core only simulations.

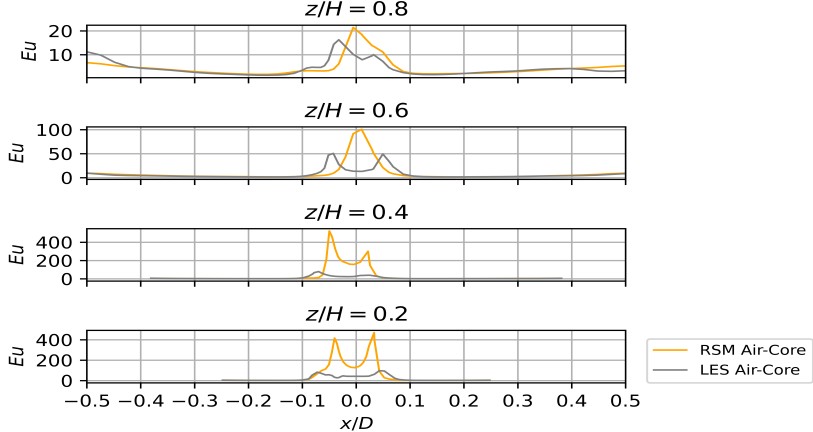

**Figure 22.** Euler Number for the air-core models as a function of the normalised $x$ co-ordinate at various heights at flow-time 2 s.

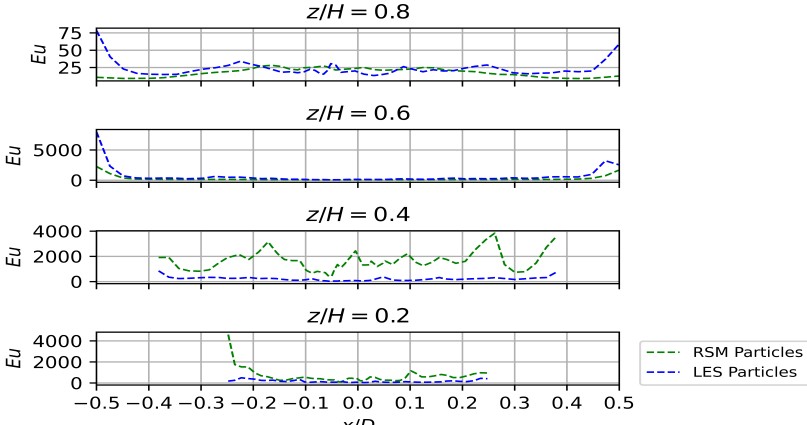

**Figure 23.** Euler Number for the particle models as a function of the normalised *x* co-ordinate at various heights at flow-time 4 s.

## 4. Conclusions

The aim in this study was to investigate the modification of the dynamic behaviour of the cyclonic flow in a hydrocyclone under surging conditions. Predictions from coupled CFD-DEM models of a hydrocyclone were implemented and benchmarked against experimental measurements. The models demonstrated good agreement with experiment; however, the LES is not suitable for coarse-grained meshes as the underflow predictions do not compare well with experiment. Furthermore, the LES model overpredicts the velocities in comparison to the RSM model. Thus, it is advised that for coarse-grained meshes, the RSM is used for modelling turbulence as it compares well with experiment and removes the need for the additional computational cost associated with fine meshes and the LES.

The phenomenological and physics-based discussion of the modification of the dynamic behaviour of the cyclonic flow before and after particles are added to the system, and surging occurs, revealed that the primary cyclonic separation mechanism breaks down during surging and results in air-core suppression. This is demonstrated via the effect of surging on the tangential and axial velocities, static and dynamic pressure, vorticity, and the Euler Number.

The following recommendations are made for future work, either to overcome some of the problems discovered in this study or to further investigate promising areas revealed in this study:

- The implementation of a more physically sound soft-contact DEM model, with particle rotation, as in [5,15], to compare the predictions to the DEM model in this study.
- The incorporation of more granular interactions via a Dense Discrete Phase Model, namely: solid shear-stress, turbulence dispersion, granular temperature, and solid pressure.
- To model larger flow times so as to capture the transient flow effects to understand the dynamics related to the potential recovery from surging and the possible/subsequent re-formation of the air-core.
- Use the understanding of the effect of surging on hydrocyclone performance to improve hydrocyclone design and operating conditions so as to operate a hydrocyclone at the highest possible feed rate without the occurrence of surging.
- It would be a valuable study to use CFD to determine the impact of contact angle (for various materials) on air-core dynamics and overall hydrocyclone dynamics and/or performance.

**Author Contributions:** Conceptualization, methodology, modelling validation, formal analysis, data curation, writing—original draft preparation, visualization, funding acquisition, project administration by M.B.; writing—review and editing by M.B., S.H.C. and A.L.N.; supervision by S.H.C. and A.L.N. All authors have read and agreed to the published version of the manuscript.

**Funding:** This research was funded by the University of Johannesburg Next-Generation Scholarship (NGS), the National Research Foundation (NRF) Doctoral Scholarship and the National Research Foundation (NRF) Thuthuka Funding Instrument (Post-PhD Track) grant number TTK160615171464.

**Data Availability Statement:** Data supporting reported results can be obtained by contacting the corresponding author.

**Acknowledgments:** The financial assistance of the National Research Foundation (NRF) and University of Johannesburg Next Generation Scholarship (UJ NGS) towards this research is hereby acknowledged. The Centre for High Performance Computing (CHPC) is acknowledged for the use of the CHPC facilities. The support and encouragement of Multotec Pty (Ltd.) in the pursuit of this research is also acknowledged. Opinions expressed and conclusions arrived at are those of the authors and are not necessarily to be attributed to the NRF, the University of Johannesburg, the CHPC or Multotec Pty (Ltd.).

**Conflicts of Interest:** The authors declare no conflict of interest.

## Abbreviations

The following abbreviations are used in this manuscript:

| | |
|---|---|
| MDPI | Multidisciplinary Digital Publishing Institute |
| DOAJ | Directory of open access journals |
| DMC | Dense medium cyclone |
| CFD | Computational fluid dynamics |
| CGD | Computational granular dynamics |
| DEM | Discrete element method |
| VOF | Volume-Of-Fluid |
| RSM | Reynold's Stress Model |
| LES | Large Eddy Simulation |
| SGS | Sub-grid scale |
| PSD | Particle size distribution |
| RANS | Reynold's Averaged Navier–Stokes |
| CSS | Continuum Surface Stress |
| DPM | Discrete phase model |
| QUICK | Quadratic Upstream Interpolation for Convective Kinematics |
| PU | Polyurethane |
| NRF | National Research Foundation |
| NGS | Next-Generation Scholarship |

## Nomenclature

| | |
|---|---|
| $a_i$ | Spherical drag law coefficient constants $\forall i = 1, 2, 3$ |
| $B_i$ | Body force components $\forall i = x, y, z$ |
| $C_D$ | Drag coefficient |
| $C_{ij}$ | Transport of the Reynolds stresses by convection |
| $C_s$ | Smagorinsky constant |
| $C_{vm}$ | Virtual mass factor |
| $D$ | Diameter |
| $D_H$ | Hydraulic diameter, m |
| $D_{L,ij}$ | Transport of the Reynolds stresses by molecular diffusion |
| $D_{T,ij}$ | Transport of the Reynolds stresses by turbulent diffusion |
| $d$ | Diameter, m |
| $\bar{d}$ | Mean diameter, m |
| $Eu$ | Euler Number, |
| $\bar{e}_{ij}$ | Unit distance vector from particle $i$ to particle $j$, m |

| | |
|---|---|
| $F_{CSS}$ | Surface tension force, N |
| $\bar{F}_{friction}$ | Friction force, N |
| $F_{ij}$ | Transport of the Reynolds stresses by production by system rotation |
| $\bar{F}_{ij}$ | Force exerted on particle $i$ by particle $j$, N |
| $\bar{F}_{other}$ | Other particle interaction forces, N |
| $G_{ii}, G_{ij}$ | Transport of the Reynolds stresses by buoyancy production |
| $\mathbf{g}$ | Gravitational vector, m/s$^2$ |
| $H$ | Hydrocyclone height, m |
| $I_{turb}$ | Turbulence intensity, % |
| $I$ | Unit tensor |
| $K$ | Spring constant, N/m |
| $k$ | Turbulence kinetic energy, J/kg |
| $L_s$ | Mixing length |
| $M_t$ | Turbulent Mach Number |
| $m$ | Mass, kg |
| $\dot{m}$ | Mass-flow rate, kg/s |
| $N$ | Number of discrete size bins |
| $n$ | Rosin–Rammler distribution spread parameter |
| $\hat{n}$ | Unit surface normal vector |
| $\bar{n}$ | Volume fraction gradient |
| $\hat{n}_w$ | Unit normal vector at the wall |
| $P$ | Pressure, kPa |
| $\Delta P$ | Pressure drop, kPa |
| $P_{ii}, P_{ij}$ | Transport of the Reynolds stresses by stress production |
| $\bar{p}$ | Mean pressure, Pa |
| $r$ | Radius, m |
| $Re$ | Reynolds Number |
| $\mathbf{S}$ | Strain-rate tensor |
| $|\mathbf{S}|$ | Tensor norm of strain-rate tensor |
| $\mathbf{S}_{ij}$ | Rate-of-strain tensor for the resolved scale |
| $S_M$ | Momentum source term |
| $S_k$ | Turbulence kinetic energy source term |
| $S_{\overline{\rho u_i' u_j'}}$ | Reynolds stresses source term |
| $t$ | Time, s |
| $\hat{t}_w$ | Unit tangential vector at the wall |
| $\Delta t$ | Time step size, s |
| $\bar{\mathbf{u}}$ | Mean velocity vector, m/s |
| $\overline{u_i}, \overline{u_j}, \overline{u_l}$ | Mean velocity components $\forall i, j, l = x, y, z$, m/s |
| $\mathbf{u}$ | Velocity vector, m/s |
| $|\mathbf{u}|$ | Velocity vector magnitude, m/s |
| $\mathbf{u}_{wall}$ | Wall velocity vector, m/s |
| $u_i', u_j', u_l'$ | Fluctuating velocity components $\forall i, j, l = x, y, z$, m/s |
| $u_i, u_j, u_l$ | Instantaneous velocity components $\forall i, j, l = x, y, z$, m/s |
| $\overline{u_i'}, \overline{u_j'}, \overline{u_l'}$ | Average fluctuating velocity components $\forall i, j, l = x, y, z$, m/s |
| $V_c$ | Computational cell volume, m$^3$ |
| $u$ | Speed, m/s |
| $v_{ij}$ | Relative velocity of arbitrary particles $i$ and $j$, m/s |
| $v_\theta$ | Tangential velocity, m/s |
| $x, y, z$ | Cartesian co-ordinates, m |
| $x_i, x_j, x_l$ | Cartesian directions $\forall i, j, l = x, y, z$, m |
| $\bar{x}_i$ | Position of particle $i$, m |
| $Y_d$ | Mass fraction of particles with diameter greater than $d$ |
| $\mathbf{0}$ | Zero vector |

**Greek Symbols**

| | |
|---|---|
| $\alpha$ | Volume fraction |
| $\gamma$ | Damping coefficient, $(\text{Ns})/\text{m}$ |
| $\delta$ | Particle overlap due to collision, m |
| $\delta_{ij}$ | Kronecker Delta |
| $\epsilon$ | Turbulence dissipation rate, $\text{m}^2/\text{s}^3$ |
| $\epsilon_{ij}$ | Transport of the Reynolds stresses by dissipation |
| $\zeta$ | A normally distributed random number |
| $\theta_w$ | Contact angle at the wall |
| $\kappa$ | von Kármán constant |
| $\mu$ | Viscosity, $\text{kg}/(\text{m s})$ |
| $\mu_f$ | Surface friction coefficient |
| $\mu_t$ | Turbulent (eddy) viscosity, $\text{kg}/(\text{m s})$ |
| $\nu$ | Kinematic viscosity, $\text{m}^2/\text{s}$ |
| $\nu_t$ | Subgrid-scale eddy viscosity, $\text{m}^2/\text{s}$ |
| $\overline{\rho u_i' u_j'}$ | Reynolds stresses (tensor notation), Pa |
| $\rho$ | Density, $\text{kg}/\text{m}^3$ |
| $\sigma$ | Surface tension coefficient, N/m |
| $\sigma_{ij}$ | Stress tensor due to molecular viscosity, Pa |
| $\sigma_k$ | Turbulent Prandtl Number for $k$ |
| $\tau_{ij}$ | Filtered sub-grid scale stress tensor, Pa |
| $\tau_p$ | Particle residence time , s |
| $\phi_{ij}$ | Transport of the Reynolds stresses by pressure strain |

**Subscripts**

| | |
|---|---|
| *air* | Air |
| *ave* | Average |
| *exp* | Experiment |
| *max* | Maximum |
| *min* | Minimum |
| *over* | Overflow |
| *p* | Particles or solids |
| *under* | Underflow |

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
