# Peer review of "The Modification of the Dynamic Behaviour of the Cyclonic Flow in a Hydrocyclone under Surging Conditions"

_mca, doi:10.3390/mca27060088_

Round 1
Reviewer 1 Report
1) Define PSD (caption table 2)
2) Did you use the same mesh sizes for LES computation. From my point of view, the given sizes of the mesh are irrelevant for LES, being too coarse.
3) Do not understand what the range of the flow rates is in Table 1. There are summations of upper and bottom limits (+1.01+0.54)/(-0.49-0.54). What are the true experimental errors?
4) Specify “different phases” L136 (is it water and solids or what?)
5) Co-efficient -> coefficient L146
6) Why contact angle 90 deg.? It strongly depends on the solid interface properties (for steel is around 78 deg.)
7) What is alpha in eq. 13?
8) The force F_CSS does not appear in momentum eq.
9) Line 220: Please rephrase “To ensure that convergence was achieved at each time step 5 - 15 Interactions”. It does not sound clear.
10) In the detail of fig. 7 it seems only a discrete cell contains air. It may easily be a numerical glitch or a reversed flow.
11) The pressure contours in Fig. 10 are indistinguishable. Choose another colormap or another scale.
12) Fig 12.: What is the time stamp for the given pressure profiles?
13) Fig. 13: Likewise.
14) Figs. 13-16: if giving the profiles with respect to a linear coordinate it would mean the flow is symmetric at that particular height. Is it true? Choosing another azimuth for Y-Z plane would you get the same results? From the given profiles it seems there is no symmetry.
15) Eq. 25: How do you correlate the Eu number definition with the profiles given in Figs. 16 and 17. In the definition you mentioned the pressure drop from the inlet to the outlets. What is the inlet and outlet for the plane section at each normalized height?
Reviewer 2 Report
This manuscript presents a VOF-DEM study of hydrocyclone. This is an interesting work. However, it must be improved for publication, as detailed below:
1) The references must be improved. The authors cited lots of papers about dense medium cyclones. However, the papers regarding hydrocyclones as considered by the manuscript are not much cited.
2) What is the difference between the present work and those reported in the paper `Modelling the multiphase flow in hydrocyclones using coarse-grained VOF-DEM and Mixture-DEM approaches`, IND ENG CHEM RES, 57(2018), 9641-9655?
3) The particle flow in the surging process should be given and demonstrated.
4) RSM and LES should give comparable results. More detail can be found in the paper " `Computational investigation of the effect of particle density on the multiphase flows and performance of hydrocyclone`, MINER ENG, 90 (2016) 55-69."
5) How many particles were simulated? Did the authors introduce coarse-grained treatment?
Round 2
Reviewer 1 Report
Accept in present form
Reviewer 2 Report
Acceptable